# OpenReview forum: "VLMGuard: Bootstrapping Malicious Prompt Detectors from Unlabeled Vision-Language Prompts in the Wild"
_TMLR — Accepted by TMLR_

### Review · Reviewer_qWXj · 2026-05-10

**Summary Of Contributions:**

The paper introduces VLMGuard, a two-stage framework for detecting malicious multimodal prompts using unlabeled VLM prompt mixtures. It extracts hidden representations from a fixed VLM, applies SVD to derive a projection-based maliciousness score, uses this score to create noisy pseudo-labels, and trains a lightweight MLP safeguard classifier for test-time detection. The paper reports strong AUROC results across three benchmark mixtures and multiple VLM backbones, with ablations on layer choice, subspace dimension, malicious ratio, transferability, and scaling.

The main strengths are the practical problem formulation, the simple and efficient representation-based method, and the broad empirical comparison. The main weaknesses are that the “unlabeled” claim depends on an unclear validation/tuning protocol, the evaluation relies on synthetic benchmark mixtures rather than real deployment logs, AUROC is insufficient for low-prevalence safety deployment, and the adaptive robustness and theoretical justification for rare-contamination settings remain limited.

**Audience:**

Yes

**Audience Explanation:**

The paper addresses an important problem in multimodal safety: detecting malicious VLM prompts before generation. The core idea—bootstrapping a detector from unlabeled mixed prompt data using VLM hidden representations—is simple, practical, and likely interesting to researchers working on VLM safety, jailbreak detection, weak supervision, and representation-based monitoring.

**Broader Impact Concerns:**

The main broader-impact concern is overblocking: the detector may conflate malicious intent with legitimate but safety-sensitive queries, especially in medical, financial, or legal domains. The use of unlabeled user prompts also raises privacy and consent concerns that deserve more discussion. The paper should address data retention, anonymization, and potential disparate false positives across languages, domains, or user groups in the appendix.

**Claims And Evidence:**

Yes

**Claims Explanation:**

The paper clearly expresses the motivation, the problem setup and the conducts detailed experiments and ablations to support the statements. It provides strong AUROC results across several benchmark mixtures, VLM backbones, and ablations, which supports the claim that VLMGuard is a promising representation-based detector. However, the strongest claims are not fully supported. The “unlabeled” claim is weakened by unclear use of a 100-sample validation set for selecting the layer, k, and threshold. The “in-the-wild” framing is also stronger than the evidence, since the evaluation uses constructed benchmark mixtures rather than real deployment logs. The paper should additionally report low-prevalence deployment metrics beyond AUROC, such as AUPRC and TPR at very low FPR.

**Requested Changes:**

1. State whether the 100-sample validation set is labeled, how it is sampled, whether it is class-balanced or drawn from the same mixture, and exactly what objective is used to select the layer, k, and threshold T. If labels are used, revise the “no human annotations” claim or provide a fully unsupervised hyperparameter-selection procedure.
2. For each dataset, backbone, and malicious ratio, report the size of the pseudo-malicious set M, the size of the pseudo-benign set B, pseudo-label precision, pseudo-label recall, and contamination rate. This is important because at π=0.005, some datasets contain only 4–6 malicious training examples.
3. Add deployment-relevant metrics. Report AUPRC under the true class prior, FPR@95%TPR, TPR@1%FPR, TPR@0.1%FPR, and confusion matrices at selected operating thresholds. AUROC alone is insufficient for a rare-event safety detector.
4. Clarify whether all baselines receive the same validation budget. In particular, Appendix A states that GradSafe uses two malicious prompts selected from the unlabeled prompt dataset; this appears inconsistent with a genuinely unlabeled setting and should be corrected or justified.
5. Add controls where benign and malicious samples are better matched by topic, prompt length, image type, source dataset, etc. This would help show that the detector is learning malicious intent rather than dataset/source artifacts.
Revise the theoretical discussion.
6. Clarify the theory for the very low malicious-rate setting. The paper argues that VLMGuard is well suited to rare malicious prompts, but the condition used to recover the SVD direction becomes harder to satisfy as the malicious fraction decreases. The authors should explain why the method still works at $\pi=0.005$, or soften the theoretical claim.
7. Strengthen adaptive attack evaluation. Report standard detection metrics under adaptive paraphrasing, include benign prompts, and evaluate stronger attacks, including multimodal perturbations, typographic prompt injection, score-query attacks, and, if appropriate, white-box attacks against the learned classifier.
8. Distinguish malicious intent, unsafe requests, regulated-domain advice, policy-sensitive requests, and benign hard negatives. Some examples appear to involve medical or financial advice, which may be safety-sensitive without necessarily being malicious.

---

### Review · Reviewer_wZBc · 2026-05-14

**Summary Of Contributions:**

The paper introduces VLMGuard, a novel two-stage framework for detecting malicious prompts in Vision-Language Models (VLMs) using unlabeled user prompts collected in the wild, without requiring human annotations.
Key contributions:
1.	Problem formulation: Formalizes malicious prompt detection as a learning problem from unlabeled mixture data following the Huber contamination model.
2.	Maliciousness estimation score: Proposes an SVD-based scoring function using VLM internal representations to estimate prompt maliciousness via projection energy onto dominant singular directions, with theoretical justification showing class-conditional separation.
3.	Safeguarding prompt classifier: Trains a lightweight MLP classifier on the noisy pseudo-partition, which provides robustness against realistic prompt variations that degrade direct projection scores.
4.	Comprehensive evaluation: Extensive experiments on multiple benchmarks with two VLM families, showing average AUROC improvement of 9.46% over SOTA.
Strengths:
•	Practical: No labeled data required; leverages naturally occurring unlabeled prompts
•	Theoretically grounded: Provides population-level analysis for SVD-based separation
•	Robust: Handles prompt variations, abnormal benign samples, and different malicious ratios
•	Scalable: Works on 72B-parameter VLMs
Weaknesses:
•	Requires access to VLM internal representations (not available for black-box APIs)
•	Performance depends on the benign/malicious mean gap and contamination ratio
•	Threshold T and number of singular vectors k require validation set tuning

**Audience:**

Yes

**Audience Explanation:**

The paper addresses a timely and important problem: safeguarding VLMs against malicious inputs. TMLR's audience (machine learning researchers interested in trustworthy AI, representation learning, and practical deployment) would find value in:
1.	The novel unsupervised learning paradigm for safety-critical detection using naturally occurring data
2.	The theoretical insight linking SVD directions to maliciousness under Huber contamination
3.	Practical implications: Deployable without labeled data, works with rare malicious examples
4.	Connections to representation engineering, outlier detection, and self-supervised learning

**Broader Impact Concerns:**

The paper already includes an Impact Statement discussing positive applications for AI safety. This is sufficient.
No additional concerns—the work is clearly aimed at defense/red-teaming to improve VLM safety. Potential dual-use concerns are inherent to all security research and are appropriately acknowledged via the "adaptive attack" experiment. The authors do not release malicious prompts or attack code beyond standard benchmarks.

**Claims And Evidence:**

Yes

**Claims Explanation:**

The evidence is strong and well-presented:
1.	Quantitative results: VLMGuard outperforms 10+ baselines across 3 scenarios, with 93.00% (LLaVA) and 94.79% (Qwen) average AUROC, improving over HiddenDetect by ~4-5% and over LLM-as-judge methods by larger margins.
2.	Ablation studies: Systematically validates design choices—layer selection, number of singular vectors, filtering threshold, embedding location.
3.	Theoretical support: Formal derivation showing expected score gap ≥ (1-π)Δ²/4 - 2σ² under spike-dominance condition.
4.	Qualitative examples: Demonstrates score alignment with malicious intent.

**Requested Changes:**

1.	Clarify the practical assumption about user consent: The paper states prompts are collected "with user consent". For real-world deployment, this is non-trivial. Please clarify: Is the framework designed for platforms that already have consent mechanisms? Or is this a limitation?
2.	Provide empirical verification of the theoretical condition: Proposition 3.3 assumes spike dominance. Please add an empirical check on your datasets to verify when the condition holds.
3.	A broader discussion: The article mentions LLM prompts, many novel articles and methods have emerged in this field at present, including Seeing Sarcasm Through Different Eyes: Analyzing Multimodal Sarcasm Perception in Large Vision-Language Models and Exploiting the Relationship within the Unlabelled Samples by Set Matching for Generalized Category Discovery.
4.	Disclose validation set composition: The validation set for tuning T, k, and layer index is "100 randomly chosen samples that are disjoint from the training and test sets". Are these samples labeled? If yes, this partially contradicts the "no human annotations" claim. Please clarify.
5.	Add standard deviations for main results Table 1: Only VLMGuard shows ± std in Table 1; other methods do not. Add standard deviations for all methods or justify omission.

---

### Review · Reviewer_Nit6 · 2026-05-19

**Summary Of Contributions:**

**Summary**

This paper proposes VLMGuard, an unsupervised malicious prompt detection framework for vision-language models. The core motivation is that obtaining large-scale labeled malicious multimodal prompts is expensive and difficult in practice. To address this challenge, the authors leverage naturally occurring unlabeled prompts collected from real-world VLM deployments, which are modeled as a mixture of benign and malicious samples under a Huber contamination formulation.

The proposed method first extracts latent prompt representations from a VLM and applies SVD to identify dominant singular directions in the embedding space. Based on these directions, the paper introduces a maliciousness estimation score that measures projection energy onto the dominant subspace, which is then used to generate noisy pseudo-labels for malicious and benign prompts. A lightweight safeguarding classifier is subsequently trained on these pseudo-labels to improve robustness under realistic prompt variations such as paraphrasing.

The paper additionally provides theoretical motivation showing that, under the Huber contamination model, malicious prompts are expected to exhibit larger projection energy along leading covariance directions when the contamination ratio is small. Extensive experiments across multiple multimodal jailbreak benchmarks and VLM backbones demonstrate strong detection performance and significant improvements over prior activation-based, uncertainty-based, and guard-model baselines.

**Strengths**
- The paper is generally well-written and easy to follow, with a clear motivation and intuitive methodology.
The idea of leveraging unlabeled in-the-wild multimodal prompts is practically important and addresses a realistic deployment challenge.
- The proposed SVD-based latent subspace analysis is simple yet effective, and the theoretical intuition under the Huber contamination model strengthens the paper.
- The empirical results consistently outperform strong prior methods, especially activation-based detectors such as HiddenDetect and ASTRA.
- The robustness analyses regarding transferability across datasets, different contamination ratios, and abnormal benign samples are valuable and strengthen the practical relevance of the work.

**Weaknesses**
- The paper lacks sufficiently strong evaluation against adaptive adversaries. Since the detection signal is derived from latent representations and dominant singular directions, an attacker may potentially optimize prompts to evade the learned subspace.
- The evaluation does not include comparisons against stronger and more recent multimodal safety systems or proprietary frontier VLMs (e.g., GPT-5 series or Claude 4.x safety systems), making it difficult to assess competitiveness against modern production-grade safeguards.
- The paper focuses primarily on activation-based detection but provides limited discussion comparing this approach against simpler text-only input classifiers or output-based moderation strategies.

**Audience:**

Yes

**Audience Explanation:**

The paper addresses an important and timely problem in AI safety and multimodal model security: detecting malicious prompts for VLMs under limited supervision. As VLMs are increasingly deployed in open-world environments and agentic systems, scalable safeguards that do not rely on expensive labeled datasets are highly relevant to both academia and industry.

The proposed approach is also technically interesting because it combines representation analysis, unsupervised learning, and practical safety evaluation in a relatively lightweight framework. Researchers working on AI safety, multimodal learning, representation analysis, jailbreak defenses, and trustworthy foundation models would likely find the paper relevant.

Moreover, the work contributes to the growing literature on activation-based safety monitoring and demonstrates that latent-space structure can provide meaningful signals for malicious intent detection even without explicit supervision.

**Broader Impact Concerns:**

The paper studies malicious prompt detection for VLMs and is generally aligned with improving AI safety and robustness. The work could help mitigate harmful multimodal jailbreak attacks and improve the reliability of deployed VLM systems. One potential concern is that the paper may also provide insights into the limitations of current multimodal safeguard systems and activation-based detectors, which could indirectly assist attackers in designing adaptive jailbreak strategies. However, the overall contribution is primarily defensive and the benefits appear to outweigh the risks. The paper already includes a brief disclaimer regarding potentially offensive examples. A more explicit broader impact discussion on dual-use considerations and adaptive attack risks would further strengthen the submission.

**Claims And Evidence:**

Yes

**Claims Explanation:**

The paper provides strong empirical evidence across multiple datasets, model families, and evaluation settings. In particular, the authors compare against a broad set of baselines spanning guard models, uncertainty-based methods, denoising-based defenses, and activation-based approaches. The experimental improvements are generally large and consistent across different scenarios. The paper also includes several useful ablation studies analyzing: (1): the role of the safeguarding classifier, (2): different transformer layers, (3): the number of singular directions, (4): different representation extraction locations, and (5): robustness under varying contamination ratios.

Additionally, the controlled perturbation experiments provide convincing evidence that the second-stage classifier improves robustness beyond directly thresholding the SVD projection score. That said, the evidence would be stronger with more comprehensive adaptive attack evaluations and broader comparisons against frontier multimodal safety systems.

**Requested Changes:**

- Add stronger adaptive attack evaluations. In particular, evaluate whether an adversary can optimize prompts to evade the SVD-based maliciousness score or the downstream safeguarding classifier while preserving attack effectiveness.
- Clarify the threat model more precisely. It would be useful to specify what level of attacker knowledge is assumed (e.g., white-box access to hidden representations, knowledge of the detector architecture, access to pseudo-labeling procedure, etc.).
- Include additional discussion on practical deployment constraints, especially regarding methods that require access to internal hidden representations of proprietary VLMs.
- Compare against stronger or newer multimodal safety systems, including frontier proprietary models if feasible.
- Add comparisons with simpler baselines such as text-only prompt classifiers or multimodal moderation APIs to better contextualize the gains from latent-space analysis.
- Discuss computational overhead and scalability in realistic deployment settings with large-scale streaming prompts.

---

### Decision · Action_Editor_eJeP · 2026-07-02

**Recommendation:** Accept as is

**Audience:**

Yes

**Audience Explanation:**

Malicious prompt detection for VLMs will be of interest to the AI safety and security community.

**Claims And Evidence:**

Yes

**Claims Explanation:**

The paper studies malicious prompt detection for vision-language models (VLMs) using unlabeled data. Specifically, it applies SVD to the internal representations of a VLM to separate unlabeled malicious and benign prompts, and then trains a binary classifier to perform detection.

All reviewers appreciated the proposed practical framework and the strong empirical performance across benchmarks. During the rebuttal, the authors provided extensive clarifications, including new experimental results on adaptive attack evaluations, comparisons with frontier closed-source multimodal safety systems, and results on confounder controls. These responses addressed most of the reviewers' concerns.

Overall, this paper introduces a simple and effective approach for detecting malicious prompts in VLMs, which is a valuable contribution to the field.